# Neighborhood Effect on Elderly Depression in Republic of Korea

**DOI:** 10.3390/ijerph20065200

**Published:** 2023-03-15

**Authors:** Ji-Hyon Hwang, Yong-Jin Kim

**Affiliations:** 1Industry-Academic Cooperation Foundation, Korea National University of Transportation, Chungju 27469, Republic of Korea; hjh860623@gmail.com; 2Division of Civil, Environmental and Urban Transportation Engineering, Korea National University of Transportation, Chungju 27469, Republic of Korea

**Keywords:** depression, neighborhood effect, spatial autocorrelation, hot spot, cluster analysis

## Abstract

This study analyzed the spatial distribution patterns of depression among vulnerable elderly across Republic of Korea. The average level of depression in the basic administrative districts was derived using the individual depression scores from the Health Interview Survey data. Results of the spatial autocorrelation analysis revealed that the Moran’s I value was 0.3138, indicating the existence of a neighborhood effect in the depression of the vulnerable older adults at the regional level. Subsequently, cluster analysis and one-way ANOVA were conducted for the hot spots where vulnerable older adult depression was concentrated. Based on the cluster analysis results, hot spots were the areas where the facilities that are necessary for the daily lives of older adults were insufficient and were categorized into three types. The findings indicate that environmental characteristics at the regional level should be considered in addition to the environmental characteristics of the house and neighborhood, which have been primarily addressed in previous studies.

## 1. Introduction

In 2020, the United Nations Department of Economic and Social Affairs [1] reported that the world’s older adult population aged 65 or older is expected to increase to over 1.5 billion by 2050, accounting for roughly 16.0% of the total population. Republic of Korea has already entered an aging society, with older adults accounting for 14.2% of its total population in 2017, and is expected to rise to 20.3% in 2025, reaching a super-aged population [2]. In old age, humans experience negative changes such as injuries, diseases, financial hardship, role and status loss, and the death of a spouse or a friend. These changes increase the vulnerability of older adults to depression [3,4,5,6,7]. Depression is the most prevalent mental health problem among older adults and occurs more frequently than in other age groups [4,8,9,10,11,12]. According to the National Health Insurance Service (2022), among the respondents who reported experiencing depression, those aged 60 or older accounted for a proportion of 22.7%, which was the second highest after those in their 40s, who accounted for 27.57% [13].

Depression negatively affects the quality of life, causes diseases, and may lead to suicide [14,15]. Therefore, considering the rapid increase in the number of older adults, depression may substantially impact society in the future, incurring huge costs in medical care and public welfare [16]. Thus, depression in older adults is not only an individual health problem but also a serious social issue that should be addressed through public policies. Ultimately, policy-makers must consider vulnerable older adults who have limited resources, such as those who live alone and suffer from poverty or other diseases, to help them manage depression.

To develop strategies to cope with depression in older adults, the characteristics of depression must be determined. Numerous studies have attempted to identify the factors affecting depression in older adults. They have emphasized that older adults’ depression is influenced more by situational factors than genetic factors [5,6,7,9,15,17,18,19]. Many of the early studies focused on individual characteristics, such as sex, age, education level, income, religious beliefs, health status, activities of daily living, and family characteristics, such as the presence of a spouse or children, presence of housemate, and relationships with family members to explain the differences in depression levels among older adults [3,9,17,20,21]. Women, older individuals, low-income people, those who live alone, those with more stressful life events, and those with less support from friends and neighbors are more likely to suffer from depression [3,4,7,9,10,17,22,23,24,25,26,27,28]. In addition, chronic illness is closely related to late-life depression [28,29]. Elderly individuals with visual impairment and circulation problems are more susceptible to depression [23]. Hypertension, coronary heart disease, diabetes, obstructive pulmonary disease, and Parkinson’s disease are frequently comorbid with depression [30,31,32]. The results are valuable in identifying specific older adult groups that are more vulnerable to depression in terms of demographic and sociological characteristics.

Researchers have discovered that depression levels vary between populations with similar individual characteristics but live in different environments. Since then, studies have been conducted to determine the relationship between the environment and depression in the older adult population. Some studies have analyzed the differences in depression levels between those who live in urban areas and those who live in rural areas. Older adults living in urban areas tend to be less depressed because they have access to various infrastructure, services, and resources [6]. Meanwhile, some studies showed opposing results where older adults living in rural areas have better mental health than older adults living in urban areas [10]. One of the proposed explanations for these results is that in rural areas, most older adults are engaged in agriculture and can work until they want to quit spontaneously without being fired. In addition, Yang [10] showed that better mental health of older adults in rural areas is associated with old communities strengthening their social support. More recent studies have shown that neighborhood characteristics that are associated with elderly depression vary between urban and rural areas [33]. Specifically, in urban areas, only nursing homes were found to be related to depressive symptoms in older adults, whereas in rural areas, the number of nursing homes, the number of social workers, and the number of senior centers were found to have a negative impact on depressive symptoms [33]. These disparities in older adults’ depression between rural and urban areas show that the environmental factors affecting depression cannot be defined simply by whether someone lives in urban or rural areas.

In subsequent studies, more specific and diverse variables have been used to describe the environmental characteristics. The environmental variables that were introduced in these studies can be classified into two categories: those that are related to socioeconomic characteristics and those that are related to physical characteristics. Socioeconomic variables include the level of crime, safety, disorder, poverty, and unemployment in a region [11,15,16]. Particularly, the social and cultural resources that are provided by local communities are recognized as important factors for mental health, as support from family, neighbors, and friends have a substantial impact on reducing depression [4,22].

Previous studies have regarded two types of physical environments that significantly impact depression in older adults. First, some studies have analyzed the relationship between depression and housing characteristics [6,34,35,36]. Second, other studies have focused on the physical environment characteristics that are formed at the neighborhood level [14,34,37,38]. These studies emphasized that the daily activities of older adults tend to be concentrated around their homes because of retirement and reduced physical ability and that the older adults are affected by the living environment more than other age groups [6,14,39,40]. Specifically, neighborhood environmental characteristics that can promote physical, leisure, and social activities that strengthen social networks are vital for the mental health of older adults [7,41].

All these studies showed that the level of depression in older adults varies in different environments and explained that depression in the older adults may appear stronger or weaker. In addition, they implied that providing individual medical treatments and creating environments that can support positive emotions is very crucial. However, empirical studies that explain the relationship between mental health and the environment in older adults remain lacking. Despite empirical evidence confirming that housing and neighborhood environment characteristics determine the mental health of older adults, limitations exist in understanding the spatial distribution patterns of depression in older adults. This finding has several reasons. First, most studies defined the range of daily life of the older adults within the neighborhood, and the environmental characteristics that were expected to affect depression in the older adults’ depression were also limited to the neighborhood level. As a result, it is difficult to identify more wide-area factors affecting the mental health of older adults. Second, several studies have focused on specific case areas. Therefore, generalizing the results of previous studies and applying them to other regions is challenging. Third, there exists a lack of consideration for the possibility of older adults’ depression forming a special aggregation pattern due to interactions between adjacent areas. For example, when polluting facilities such as factories and thermal power plants are located in a particular area, all individuals residing in the adjacent neighborhoods may be adversely affected by the facilities, resulting in a concentration of poor health people around the area.

Most geographic phenomena occur intensively around a specific location and form a certain spatial pattern. Tobler [42] presented the first law of geography where “Everything is related to everything else, but near things are more related than distant things”. This means that the phenomenon that occurs in a space is caused by the unique characteristics of the space itself but is also affected by other spaces that are adjacent to the space. Thus, regions with similar values for a particular phenomenon are adjacent to each other, called spatial autocorrelation.

Although depression in older adults in Republic of Korea may form a particular spatial pattern, finding studies analyzing depression in older adults at the regional level is difficult, considering the possibility of spatial autocorrelation. As mentioned above, understanding depression in older adults as a social problem that is caused by the surrounding environment rather than individual problems is necessary. Thus, demonstrating whether depression in older adults occurs intensively in a specific area and identifying the characteristics of these areas are essential for establishing health and welfare policies and urban policies.

Against this background, this study aimed to analyze regional differences and spatial distribution patterns of depression in vulnerable older adults in Republic of Korea and provide reference that can be used to establish policies to reduce depression in older adults. Specifically, using spatial autocorrelation analysis on the depression levels of vulnerable older adults, this study attempted to confirm whether the occurrence of depression in older adults is spatially clustered. Subsequently, specific areas where older adults’ depression is concentrated are derived from hot spot analysis, and the environmental characteristics of these areas were analyzed. This study confirmed the neighborhood effect on older adults’ depression at the regional level and provided a basis for the argument that multilateral approaches should be taken together from the urban planning perspective in addition to the previously emphasized individual-level efforts to reduce depression in older adults.

## 2. Materials and Methods

### 2.1. Data

In this study, the data of the older adults aged 65 or older from the Health Interview Survey, which was conducted as part of the 2015 Visiting Health Management Project, were used. The Visiting Health Management Project is a project that is promoted by the Ministry of Health and Welfare, one of the Korean government departments, to provide healthcare services to people who have difficulty using and accessing facilities, such as hospitals and public health centers, despite high health risk factors, such as old age, poverty, disease, and disability. The Health Interview Survey is one of the programs constituting the Visiting Health Management Project and is conducted to identify respondents’ health behaviors and health risk factors. Therefore, the subjects of this study are older adults who participated in Visiting Health Management Projects who require better health management and should be subject to seniority-related policies first.

The Short-form Geriatric Depression Scale (SGDS-K) was used to measure depression levels in the Health Interview Survey. The SGDS-K is a Korean translation of the Short-Form Geriatric Depression Scale (SGDS) that was developed by Sheikh and Yesavage [43]. This survey paper is comprised of 15 questions that must be responded with “Yes” or “No,” and the total score is calculated by giving one point for each question. If the total score is 5 points or less, it is normal, and if the total score is 6 to 9, it is judged that moderate depression exists, and if the score is 10 or higher, it is judged that depression exists, and the closer the score is to 15 points, the higher the depression level.

Data from 534,598 older adults were used in this study, excluding the data for 34,478 responses whose address information was omitted or whose depression measurement questions were not answered among a total of 569,076 older adults that were surveyed in 2015.

### 2.2. Study Area

In this study, to analyze the spatial pattern of depression in the older adults, the value that was calculated by averaging the individual depression scores in each administrative district was used as a variable to indicate the depression level by region. The administrative divisions of Republic of Korea are comprised of three hierarchies. The entire territory of Republic of Korea is comprised of Metropolitan City called Si and Province called Do as first-level administrative divisions (Figure 1a). Province and Metropolitan city are divided into Si, Gun, and Gu as second-level administrative divisions (Figure 1b). Si, Gun, and Gu correspond to city, county, and district in the United States. In addition, Si, Gun, and Gu are divided into Eup, Myeon, and Dong, the most basic administrative divisions (Figure 1c). Eup, Myeon, and Dong correspond to town, township, and neighborhood in the Unites States. They have different roles and responsibilities. Eup is larger than Myeon and is mostly located in rural areas. Myeon is smaller than Eup, and it is also located in rural areas. Dong is the smallest administrative division, and it is mostly located in urban areas.

The units of analysis in this study were Eup, Myeon, and Dongs. In 2015, there were 5039 Eup, Myeon, and Dongs in Republic of Korea. Analysis was conducted on 4114 Eup, Myeon, and Dongs, excluding 925 Eup, Myeon, and Dongs, where the Health Interview Survey was not conducted.

### 2.3. Statistical Analysis

In this study, Moran’s I and Getis–Ord’s Gi* indices were used to identify the spatial autocorrelation of depression in older adults. Moran’s I indicates whether a specific phenomenon tends to be spatially or randomly clustered. Moran’s I has a value between −1 and 1, the closer it is to 1, the more positive the autocorrelation, which means that the areas with similar values are spatially clustered, and the closer it is to −1, the more negative the autocorrelation, which means that areas with large values and areas with small values are regularly distributed. However, if Moran’s I has a value close to 0, then no spatial autocorrelation exists.

Moran’s I was used to examine whether clusters occurred in the entire study area. However, a limitation exists in not being able to provide information on which specific areas are clustered, and if they are clustered, whether they are clustered around large or small values. Accordingly, Getis–Ord’s Gi* index was used to understand local spatial autocorrelation. The Getis–Ord Gi* index was used to derive hot spots where areas with large values are clustered. The software which was used for spatial autocorrelation analysis was GeoDa(version 1.12.0).

Subsequently, hot spot areas were categorized and the characteristics of each type were identified. Cluster analysis and one-way ANOVA were employed using the two-stage methodology that was proposed by Hair and Black [44]. Cluster analysis is a method of grouping observations on the basis of the similarity of values of various variables that are related to a specific subject [45] and is consistent with the purpose of classifying hot spots of depression in the older adults. Generally, studies using cluster analysis [45,46,47] present the results in the form of a visual map describing the characteristics of cities corresponding to each cluster. In this study, cluster analysis, a method of categorization that has been used in several previous studies, was adopted to categorize the particular areas where older adults’ depression is high and to identify the characteristics of each cluster. In the cluster analysis, variables, including the average slope, the number of subway lines, road ratio, number of parks, number of traditional markets, number of hypermarkets and department stores, number of convenience stores, number of senior community centers, number of senior welfare centers, number of health centers, number of clinics, number of general hospitals, population, aged population ratio, and the percentage of buildings over 20 years in the area, were considered. Standardized Z values for each variable were calculated and used to control for the effect of different scales between the variables. A one-way ANOVA was used to verify whether significant differences existed between the clusters in each variable.

## 3. Results

### 3.1. Depression Level of Vulnerable Older Adults by Region in Republic of Korea

Table 1 presents the respondents’ basic statistics. First, out of 534,598 respondents, 137,463 (26%) were male and 397,135 (74%) were female. The average age of the respondents was 77.1 years. Approximately 53% of the respondents were in their 70s, followed by those in their 80s (approximately 31%), 60s (approximately 13%), and 90s (approximately 3%).

The average level of depression for all the respondents was approximately 4.57 out of 15. Furthermore, the average level of depression for each group in terms of sex and age was between 4 and 5 points. Thus, the level of depression in older adults is not high overall in Republic of Korea, and that no significant difference exists in the level of depression depending on sex or age.

However, as seen in Table 2, nearly 8.1% of all respondents had a depression level of 10 or more, and approximately 1.7% of all respondents had the highest depression level of 15, which means that depression is high around a small number of specific groups.

Table 3 shows the basic statistics for the depression levels of the older adults at the Eup, Myeon, and Dong levels by averaging the individual data. The survey was conducted in 4114 areas. Among them, 2934 (71.3%) were in the Dong area, and 1180 (28.7%) were in the Eup and Myeon areas. The average depression level for 4114 Eup, Myeon, and Dongs was 4.77, the average depression level for Dongs was 4.68, and the average depression level for Eup and Myeons was 4.97.

Table 4 reveals that Eup, Myeon, and Dongs with depression levels of more than five points accounted for 38.7%, and Eup, Myeon, and Dongs with more than 10 points accounted for 3.3%.

Following the previous analysis, the depression level in the older adults in Eup, Myeon, and Dongs was averaged by 17 metropolitan administrative districts, Province and Metropolitan City. Figure 2 and Table 5 show that the average level of depression in the older adults in Province and Metropolitan City was not relatively high, between 4 and 5 points. However, the proportion of Eup, Myeon, and Dongs with depression scores of 10 or more differed in metropolitan administrative districts. The top five metropolitan administrative districts with the highest percentage of Eup, Myeon, Dongs, which have a depression score of 10 points or more, were found to be Jeollanam Province (22, 6.84%), Jeollabuk Province (21, 6.84%), Chungcheongbuk Province (9, 4.57%), Chungcheongnam Province (11, 4.7%), and Gwangju Metropolitan City (5, 4.1%). Contrastingly, the top five metropolitan administrative districts with the lowest percentage of Eup, Myeon, and Dongs, which had depression scores of 10 points or more, were Ulsan Metropolitan City (0, 0.00%), Incheon Metropolitan Ccity (0, 0.00%), Sejong Special Self-governing City (0, 0.00%), Seoul Special City (3, 0.75%), and Daejeon Metropolitan City (1, 0.86%).

### 3.2. Spatial Patterns of Older Adults’ Depression in Republic of Korea

Global spatial autocorrelation analysis was conducted on 4114 Eup, Myeon, and Dongs to confirm whether depression in the older adults is concentrated around specific areas. The spatial weight for this analysis was based on the first-order Queen’s contiguity rule. The Moran’s I statistic was 0.313775, and the z-score was 26.92, confirming the spatial autocorrelation of depression in older adults. This shows that differences exist in the depression level in older adults by area, and a certain spatial pattern is formed. This confirms that older adults’ depression in each area can be affected by adjacent areas in addition to the conditions provided by the area.

Subsequently, hot spots were derived using Gi* statistics to determine specific areas where depression in older adults is specifically concentrated. A total of 313 Eup-Myeon-Dongs defined as hot spots can be seen in Figure 3.

Table 6 shows that the number of hot spots for depression in older adults was found to be high in the following order: Jeollabuk Province (16.94%), Gangwon Province (12.45%), Gyeonggi Province (11.43%), Jeollanam Province (9.97%), Chungcheongnam Province (9.13%), Daegu Metropolitan City (8.02%), Busan Metropolitan City (6.52%), Incheon Metropolitan City (5.63%), Gwangju Metropolitan City (4.92%), Gyeongsangnam Province (4.61%), Gyeongsangbuk Province (4.39%), Daejeon Metropolitan City (4.31%), Chungcheongbuk Province (4.06%), Ulsan Metropolitan City (2.94%), Jeju Special Self–governing Province (2.86%), Seoul Special City (1.25%), and Sejong Special Self–governing City (0.00%).

Table 7 shows the distribution of hot spots by urban area, rural area, and urban-rural complex area. In this analysis, the urban area was defined as an area that was composed of only Dongs among basic administrative districts, the urban-rural complex area was defined as an area where Dongs, Eup, and Myeon existed together, and rural areas were defined as an area composed only of Eup and Myeon. According to this analysis, the hot spot ratio was high in rural areas and urban-rural complex areas.

Number of hot spots by large cities and small- and medium-sized cities. On the basis of the size of the population of urban areas and urban-rural complex areas, cases with 500,000 or more were classified as large cities, and cases with less than 500,000 were classified as small- and medium-sized cities. Table 8 shows that the hot spots were concentrated in small- and medium-sized cities.

The results of this analysis indicate that depression in older adults has a neighborhood effect in a specific area. Kim [48] argued that mental health can be affected by physical activity, emotional support, opportunities for social interaction, and physical environmental characteristics. However, as Kim pointed out, studies on neighborhood effects were relatively insufficient when compared to existing studies on various causes of depression at the individual level, and few studies have explored the physical environment, especially services and convenience facilities, and the results of this study are vital in this regard.

### 3.3. Environmental Characteristics of Areas Where Older Adult Depression Is Concentrated

The results of previous studies demonstrated an actual neighborhood effect of depression in older adults at the national level. Particularly, analysis of these regions can indicate which regional characteristics are related to the spatial concentration of depression in older adults; therefore, environmental management at the regional level can improve mental health for older adults or ease the burden on older adults’ depression.

In this context, a hierarchical cluster analysis was conducted focusing on the areas where depression in older adults is concentrated to understand the environmental characteristics of these areas. After the shortest connection, the longest connection, the average connection, and the Ward methods were all performed, the Ward method, which most clearly showed cluster formation, was adopted.

As a result of cluster analysis, Table 9 exhibits three clusters that are visible in older adults’ depression-intensive area (hot spots). Of the 310 samples, excluding the three outliers that do not belong to any cluster, 170 were found to be included in Type 1, 22 were found in Type 2, and 118 were found to be included in Type 3.

Table 10 presents the results of the one-way ANOVA for the three clusters. For all the variables, the difference between the clusters was statistically significant.

Type 1 mainly includes large and small- and medium-sized cities. The shopping environments in these areas were poor, and facilities for older adults were underequipped. Although the buildings were not old, public transportation or parks were relatively insufficient in these areas. These results are similar to those of existing studies, which explained that shopping facilities or welfare facilities for older adults in the neighborhood positively affect their health. However, the space ranges of the depression-intensive areas that were derived in this study are 5 to 10 min by car, which means that shopping and welfare facilities outside the neighborhood or walking range are also important for older adults’ mental health.

In Type 2, large cities, small- and medium-sized cities, and rural areas were all included. These areas have steep slopes, the highest proportion of old buildings, and a high proportion of older adult residents. Contrastingly, infrastructure, such as public transportation facilities, shopping facilities, and parks, are well equipped. These results demonstrate that, despite the good environment, the mental health of older adults may suffer if the socioeconomic environment is poor. This is because the high proportion of old buildings in an area is related to the low-income level and security concerns. Thus, the economic level of the areas that were included in the second type was relatively low. In addition, the mental health of older adults may decrease if urban vitality decreases because of the high proportion of older adults among residents.

Finally, Type 3 includes areas with high slopes, inconvenient public transportation, and poor shopping and medical facilities. Particularly, all areas belonging to Type 3 were rural. Rural areas tend to have insufficient convenience facilities compared with urban areas. The steep slopes indicate that the areas are unsuitable for walking, public transportation is inconvenient, and the outdoor physical activity of older adults may be restricted. In addition, if shopping and medical facilities are insufficient, older adults will inevitably encounter difficulties in their daily lives. Ultimately, if areas with these characteristics are geographically adjacent, they will negatively impact the mental health of older adults because of the long distances that they must travel to shop or visit the hospital.

## 4. Conclusions

This study demonstrated the neighborhood effect on the mental health of vulnerable older adults at the regional level and argued that previous personal treatment and collective efforts in terms of urban planning must be combined to improve the mental health of older adults and prepare for a rapid transition to a super-aged society.

This study used survey data from 560,976 interview subjects that were aged 65 or older in the 2015 Visiting Health Management Project. This study observed a neighborhood effect in the depression of older adults, and 313 regions (hot spots) appeared to be concentrated in vulnerable older adults with a high level of depression. This basic evidence indicates that the mental health of vulnerable older adults is affected by various regional environmental characteristics, such as physical, social, economic, and cultural, in a situation where individual physical health status and social and economic conditions are controlled.

As a result of the cluster analysis of older adults’ depression hot spot areas, three clusters were identified. Type 1 included large and small- and medium-sized cities where shopping facilities, older adults facilities, public transportation facilities, and parks are insufficient. Type 2 includes areas with high slopes, old buildings, and a high proportion of the older adult population. The third type is rural areas with high slopes, inconvenient public transportation, and poor shopping and medical facilities.

These results can be understood in the same context as those of previous studies which showed that the mental health of older adults is affected by the surrounding environment in addition to personal conditions. Furthermore, in line with previous studies, this study confirmed that environmental characteristics that are formed at a wider regional level beyond housing or neighborhood affect the mental health of older adults and that a neighborhood effect exists, indicating that depression in older adults is spatially concentrated. This means that if it is difficult to use shopping facilities, welfare facilities for older adults, and medical facilities within the range of 5 to 10 min by car outside the neighborhood, it negatively affects the mental health of older adults in the region. In addition, this study verifies that, even if the physical environment is good, the mental health of older adults may suffer if the socioeconomic characteristics of the region are poor.

Thus, even if it is difficult to use shopping facilities, welfare facilities for older adults, and medical facilities within the range of 5 to 10 min by car outside the neighborhood, it negatively affects the mental health of older adults in the region. Moreover, this study showed that even if public transportation and the physical environment is good, such as shopping environment or park, socioeconomic characteristics, such as the overall backwardness of the region and the high proportion of older adults, can be negative for the mental health of older adults.

As most existing research and policies related to mental health for older adults suggest individual-targeted solutions, this study revealed the legitimacy of collective efforts at the local level. Particularly, although studies that revealed the relationship between depression in older adults and the environment have mainly focused on environmental characteristics within the walking area, this study confirmed that environmental characteristics at the regional level affect older adults’ depression to form a depression-intensive area.

Although this study confirmed the neighborhood effect on the mental health of vulnerable older adults, it failed to provide a detailed analysis of the areas with good and poor mental health. In-depth follow-up research in these areas is required to present alternatives at a more targeted urban planning and policy level.

## Figures and Tables

**Figure 1 ijerph-20-05200-f001:**
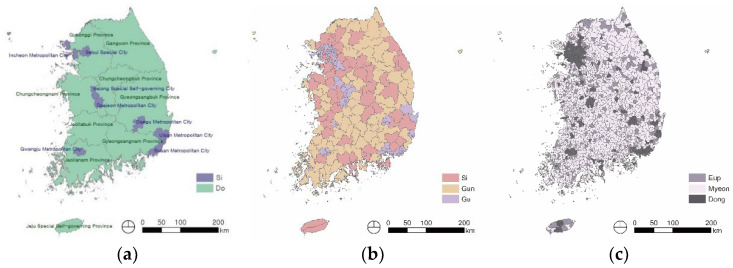
Administrative divisions: (**a**) high level, (**b**) middle-level, and (**c**) low-level.

**Figure 2 ijerph-20-05200-f002:**
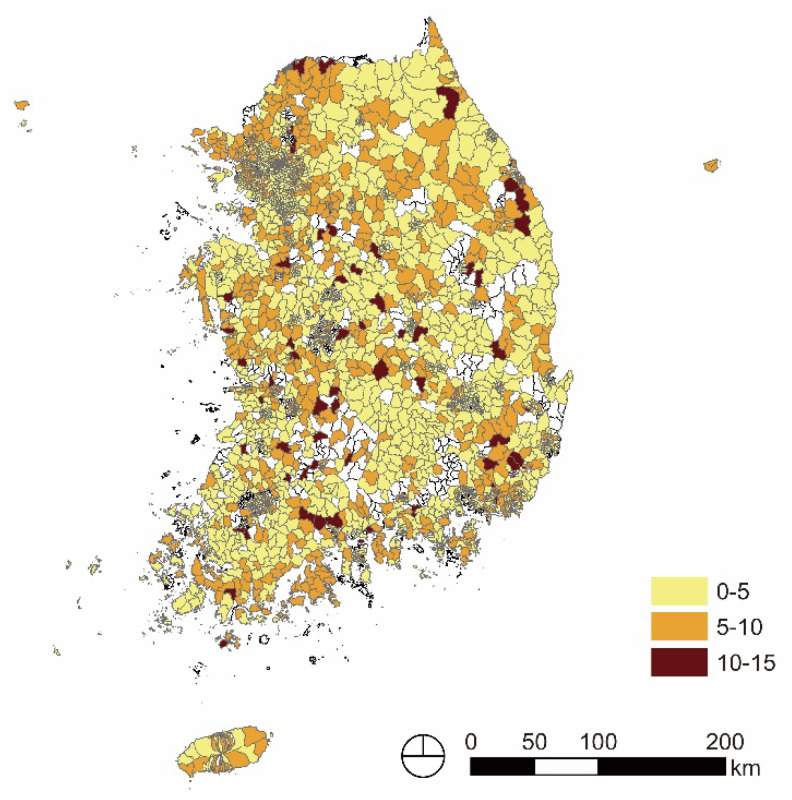
Level of older adult depression in Eup, Myeon, and Dongs.

**Figure 3 ijerph-20-05200-f003:**
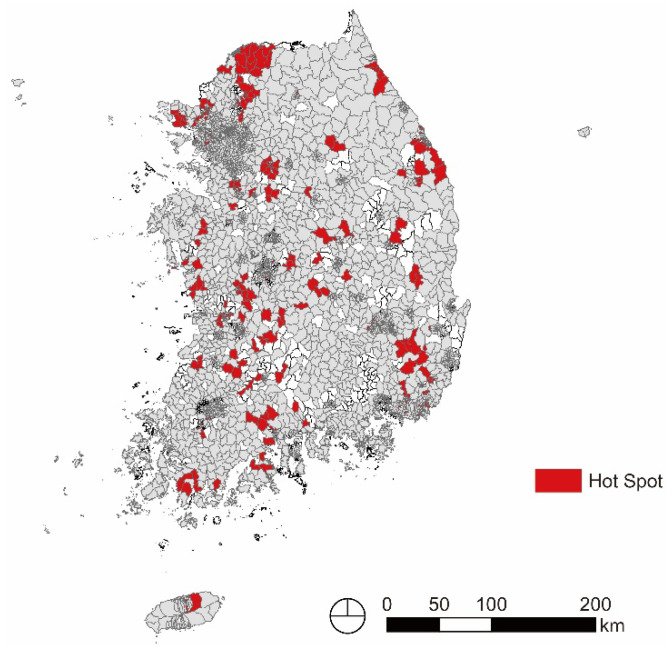
Hot spot of older adults’ depression.

**Table 1 ijerph-20-05200-t001:** Basic statistics of the individual respondents.

Group		Number of Respondents	Depression Level	
			Mean	Standard Deviation
Sex	Male	137,463	4.36	3.25
	Female	397,135	4.64	3.18
Age	60s	68,124	4.17	3.44
	70s	281,371	4.50	3.17
	80s	167,747	4.79	3.12
	Over 90s	17,356	5.09	3.21
Total		534,598	4.57	3.20

**Table 2 ijerph-20-05200-t002:** Number of individual respondents at each depression levels.

Depression Level	0–5	6–9	10–15	Total
Number of respondents	376,793 (70.48%)	114,683 (21.45%)	43,122 (8.07%)	534,598 (100.00%)

**Table 3 ijerph-20-05200-t003:** Basic statistics of the Eup, Myeon, and Dongs.

Spatial Units	Number of Spatial Units	Depression Level	
		Mean	Standard Deviation
Eup and Myeon	1180	4.97	2.64
Dong	2934	4.68	2.20
Total	4114	4.77	2.34

**Table 4 ijerph-20-05200-t004:** Number of Eup, Myeon, and Dongs at each depression level.

Depression Level	≥0 and ≤5	>5 and <10	≥10 and ≤15	Total
Number of spatial units	2522 (61.30%)	1458 (35.44%)	134 (3.26%)	4114 (100.00%)

**Table 5 ijerph-20-05200-t005:** Average level of older adult depression in Eup, Myeon, and Dongs by Province and Metropolitan City.

Province and Metropolitan City	Average Level of Older Adults Depression		Number of Eup, Myeon, and Dongs with a Depression Score of 10 or More		Total Number of Eup, Myeon, and Dongs	
	Mean	St. Dev	N	%	N	%
Seoul Special City	4.59	1.54	3	0.75	400	100.00
Busan Metropolitan City	4.57	2.46	4	2.17	184	100.00
Daegu Metropolitan City	5.05	2.21	3	1.85	162	100.00
Gwangju Metropolitan City	5.00	2.22	5	4.10	122	100.00
Incheon Metropolitan City	5.23	1.42	0	0.00	142	100.00
Daejeon Metropolitan City	4.80	1.98	1	0.86	116	100.00
Ulsan Metropolitan City	4.99	1.16	0	0.00	68	100.00
Sejong Special Self–governing City	4.48	1.28	0	0.00	15	100.00
Gyeonggi Province	5.13	2.07	19	2.90	656	100.00
Gangwon Province	4.73	2.71	9	3.50	257	100.00
Chungcheongnam Province	4.63	2.48	11	4.37	252	100.00
Chungcheongbuk Province	4.15	2.31	9	4.57	197	100.00
Jeollanam Province	4.75	3.22	22	6.85	321	100.00
Jeollabuk Province	4.89	3.07	21	6.84	307	100.00
Gyeongsangnam Province	4.81	2.13	13	3.16	412	100.00
Gyeongsangbuk Province	4.26	2.29	13	3.00	433	100.00
Jeju Special Self–governing Province	5.52	1.25	1	1.43	70	100.00
Total	4.77	2.34	134	3.26	4114	100.00

**Table 6 ijerph-20-05200-t006:** Number of Eup, Myeon, and Dongs defined as a hot spot in Province and Metropolitan City.

Province and Metropolitan City	Number of Eup, Myeon, and Dongs Defined as a Hot Spot		Total Number of Eup, Myeon, and Dongs	
	N	%	N	%
Seoul Special City	5	1.25	400	100.00
Busan Metropolitan City	12	6.52	184	100.00
Daegu Metropolitan City	13	8.02	162	100.00
Gwangju Metropolitan City	6	4.92	122	100.00
Incheon Metropolitan City	8	5.63	142	100.00
Daejeon Metropolitan City	5	4.31	116	100.00
Ulsan Metropolitan City	2	2.94	68	100.00
Sejong Special Self–governing City	0	0.00	15	100.00
Gyeonggi Province	75	11.43	656	100.00
Gangwon Province	32	12.45	257	100.00
Chungcheongnam Province	23	9.13	252	100.00
Chungcheongbuk Province	8	4.06	197	100.00
Jeollanam Province	32	9.97	321	100.00
Jeollabuk Province	52	16.94	307	100.00
Gyeongsangnam Province	19	4.61	412	100.00
Gyeongsangbuk Province	19	4.39	433	100.00
Jeju Special Self–governing Province	2	2.86	70	100.00
Total	313	7.61	4114	100.00

**Table 7 ijerph-20-05200-t007:** Number of Eup, Myeon, and Dongs defined as a hot spot in urban areas and rural areas.

Spatial Units	Number of Eup, Myeon, and Dongs Defined as a Hot Spot		Total Number of Eup, Myeon, and Dongs
	N	%	N
Urban areas	45	4.06	1109
Urban-rural complex areas	209	8.89	2352
Rural areas	59	9.04	653
Total	313	7.61	4114

**Table 8 ijerph-20-05200-t008:** Number of Eup, Myeon, and Dongs defined as a hot spot in large cities and small- and medium-sized cities.

Spatial Units	Number of Eup, Myeon, and Dongs Defined as a Hot Spot		Total Number of Eup, Myeon, and Dongs
	N	%	N
Large cities	74	3.94	1878
Small- and medium-sized cities	180	11.37	1583
Rural areas	59	9.04	653
Total	313	7.61	4114

**Table 9 ijerph-20-05200-t009:** Type of older adults depression hot spots (cluster analysis results).

Cluster	City Including Hot Spots		Number of Hot Spots
	City Type	City Name	
Type 1	Large cities	Seoul Special city, Gwangju Metropolitan city, Daegu Metropolitan city, Daejeon Metropolitan city, Busan Metropolitan city, Ulsan Metropolitan city, Goyang-Si, Changwon-Si, Pohang-Si	42
	Small- and medium-sized cities	Donghae-Si, Samcheok-Si, Chuncheon-Si, Taebaek-Si, Bucheon-Si, Yangju-Si, Icheon-Si, Pyeongtaek-Si, Pocheon-Si, Gyeongju-Si, Mungyeong-Si, Andong-Si, Naju-Si, Mokpo-Si, Suncheon-Si, Namwon-Si, Iksan-Si, Jeongeup-Si, Jeju-Si, Nonsan-Si, Asan-Si, Jeonju-Si	123
	Rural areas	Namyangju-Si, Jeongseon-Gun, Cheorwon-Gun, Yeoncheon-Gun	5
Type 2	Large cities	Daegu Metropolitan city, Busan Metropolitan city, Incheon Metropolitan city	6
	Small- and medium-sized cities	Pyeongtaek-Si, Pocheon-Si, Gimhae-Si, Iksan-Si, Jeonju-Si	9
	Rural areas	Gimpo-Si, Namyangju-Si, Icheon-Si, Paju-Si, Haenam-Gun, Wanju-Gun, Hongseong-Gun	7
Type 3	Large cities	Daegu Metropolitan city, Busan Metropolitan city	2
	Small- and medium-sized cities		0
	Rural areas	Samcheok-Si, Anseong-Si, Icheon-Si, Paju-Si, Pyeongtaek-Si, Pocheon-Si, Gimhae-Si, Miryang-Si, Yangsan-Si, Gumi-Si, Gimcheon-Si, Mungyeong-Si, Sangju-Si, Andong-Si, Naju-Si, Suncheon-Si, Gimje-Si, Namwon-Si, Iksan-Si, Jeongeup-Si, Jeju-Si, Nonsan-Si, Boryeong-Si, Asan-Si, Chungju-Si, Yangyang-Gun, Cheorwon-Gun, Hoengseong-Gun, Yeoncheon-Gun, Hadong-Gun, Cheongdo-Gun, Cheongsong-Gun, Ulju-Gun, Ganghwa-Gun, Goheung-Gun, Gokseong-Gun, Haenam-Gun, Gochang-Gun, Muju-Gun, Buan-Gun, Sunchang-Gun, Wanju-Gun, Imsil-Gun, Jinan-Gun, Buyeo-Gun, Seocheon-Gun, Yesan-Gun, Boeun-Gun, Yeongdong-Gun, Okcheon-Gun, Eumseong-Gun, Jincheon-Gun	116

**Table 10 ijerph-20-05200-t010:** Environmental characteristics of clusters (ANOVA results).

Variables	MST	df	MSE	df	F	*p*-Value
Average Slope	21.255	2	0.871	307	24.405	0.000
Number of Subway Line	18.993	2	0.734	307	25.864	0.000
Road Ratio	106.779	2	0.319	307	334.696	0.000
Number of Traditional Markets	24.614	2	0.716	307	34.380	0.000
Number of Hypermarkets	3.509	2	0.127	307	27.623	0.000
Number of convenience Stores	60.714	2	0.458	307	132.674	0.000
Number of Parks	45.048	2	0.680	307	66.213	0.000
Number of Senior Community Centers	100.479	2	0.311	307	323.511	0.000
Number of Senior Welfare Centers	10.777	2	0.755	307	14.277	0.000
Number of Health Centers	19.284	2	0.826	307	23.356	0.000
Number of General Hospitals	18.707	2	0.655	307	28.554	0.000
Number of Clinics	20.330	2	0.336	307	60.488	0.000
Population	55.602	2	0.348	307	159.782	0.000
Aged Population Ratio	45.813	2	0.705	307	65.018	0.000
Percentage of Buildings over 20 Years	67.974	2	0.457	307	148.831	0.000

## Data Availability

Not applicable.

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
