# Peer review of "Neighborhood Effect on Elderly Depression in Republic of Korea"

_ijerph, 2023, doi:10.3390/ijerph20065200_

Round 1
Reviewer 1 Report
This study is judged to have great significance in that it conducted a spatial analysis on the elderly's depression on a national basis..
I present a reading opinion on the researcher's valuable research paper.
1. Table 2 and Table 4 need to modify the notation method and result value description. Please add percentages to the table.
In the case of the result value description, please indicate 5 to 10 points for a depression index of 5 or more and 10 to 15 points for a depression index of 10 or more. Index ranges overlap.
2. In Table 5, the English name of ‘Si and Do’ needs to be modified.
example) Seoul-Si ==> Seoul Metropolitan
Author Response
Thank you for your kind comments and suggestions.
Please see the attachment.

Reviewer 2 Report
Thank you for giving me the opportunity to revise this paper.
-Introduction: the first part is well written and depicts the background of the impact of aging on population epidemiology in South Korea. Authors also introduced depression as the commonest mental health problems in aged societies. I have some comments to improve the overall quality of the manuscript.
a) At line 42-43, I suggest to add some epidemiological data on prevalence and incidence of depression in older individuals along with demographical factors (sex and age) that significantly affect prevalence of this disorder.
b) I suggest to rework lines 61-63: indeed, it is not clear the meaning of the phrase "older adults" as you are listing factors associated with depression among older individuals only. Maybe, you should change it by simply writing "increasing age", or specifying which age groups among elderly are more prone to development of depression. Moreover, the part that "people with disability or illnesses , people with low levels of daily mobile activities" should be more specific; indeed, the concept of disability often overlaps with that of low levels of daily mobile activities, so you should specify which disability was associated with depression (physical? cognitive? both?); and also, among illnesses, have specific illnesses or number of illnesses been found associated with depression?
c) Rural vs Urban areas: introduce this recently published study https://doi.org/10.3390/healthcare11040476.
d) I feel that the introduction is too long and sometimes redundant. I suggest to condensate some concepts and be as more specific as possible (e.g. explicitating diseases or disabilities associated with depression).
-Methodology and Statistical analyses are well performed and described.
-Discussion: is clearly presented and supported by references.
-Minor changes:
a) Line 61: change "low-income people" with "low income".
b) Line 71: change "de-pressed" with "depressed".
Author Response

(The authors gave the same response as above.)

Reviewer 3 Report
The work is very interesting and gives an innovative approach to the problem of depression.
I have noticed some points in the manuscript where the authors can intervene for better reader understanding:
Line 113-114, an example could better explain the meaning of the sentence
Line 116 the sentence is a bit tricky to understand, even if it is later explained by the authors. Perhaps we should add a few more elements from the point of view of the "geographic" theory
178-180 the authors should provide the translation of Eup (town) etc. (Meyon, Dongs)
Author Response

(The authors gave the same response as above.)
